# Attitudes of women towards intimate partner violence in Guyana: A cross-sectional analytical study

Gary Joseph [1,2]*, Luis Paulo Vidaletti [3], Cona Husbands[4], Lisa Edwards[5], Michelle K. James[2], Charles C. Branas[6], Christopher N. Morrison[6,7]

1 Institute of Health Science Education, Georgetown Public Hospital Corporation, Georgetown, Guyana, 2 College of Medical Sciences, University of Guyana, Georgetown, Guyana, 3 International Center for Equity in Health, Federal University of Pelotas, Pelotas, Brazil, 4 Sexual Offences and Domestic Violence Policy Unit, Ministry of Human Services & Social Security, Georgetown, Guyana, 5 Institute of Gender Studies, University of Guyana, Georgetown, Guyana, 6 Department of Epidemiology, Columbia University Mailman School of Public Health, New York, NY, United States of America, 7 Department of Epidemiology and Preventive Medicine, School of Public Health and Preventive Medicine, Monash University, Melbourne, Australia

* garyjoseph056@gmail.com

**Data Availability Statement:** The study incorporates all relevant data within the paper. Additional details on the prevalence of women's attitudes toward intimate partner violence,

## Abstract

### Background

To assess the attitudes of women towards intimate partner violence (IPV) in Guyana.

### Methods

We used national data from the publicly available Multiple Indicator Cluster Survey (MICS) conducted in Guyana in 2019 for women aged 15 to 49 years. The prevalence of women who agreed that a husband is justified in beating his wife was analyzed. Respondent reasons included if she: "goes out without telling him", "neglects the children", "argues with him", "refuses sex with him", "burns the food", "has another partner", "stays out late/partying", "refuses to cook or clean", "overspends", and/or "he doesn't have access to her cellphone". Descriptive analyses were carried for all the variables. Logistic regression was used to identify factors associated with these 10 respondent reasons, separately and in combination.

### Results

The overall prevalence of women's attitudes justifying IPV against women if there was a 'yes' response to any of the 10 reasons was 17.9% (95%CI: 16.6–19.3%), and varied from 2.7% if she "goes out without telling him", "burns the food", or "overspends" to 10.0% if she "has another partner". This prevalence ranged from 10.2% in urban areas to 19.3% in rural areas (p<0.001), and from 16.1% in coastal to 30.1% in interior areas (p<0.001). Similarly, 25.9% of women from the poorest household agreed that a husband has the right in beating his wife for any of the 10 reasons compared to 11.6% of the richest women (11.6%) (p<0.001). Rural place of residence, ethnicity, geographic region, level of education, wealth

categorized by various factors, along with results from the multivariate logistic analysis, can be found in the appendices. The data used in this study are hosted by the Multiple Indicator Cluster Survey (MICS) website at https://mics.unicef.org/. These datasets are freely available online for research purposes, and users can access the original data upon registering as MICS data users.

**Funding:** The study was funded by the National Institutes of Health, Fogarty International Center grant number D43TW012189. The funders had no role in study design, data collection and analysis, decision to publish, or preparation of the manuscript.

**Competing interests:** The authors have declared that no competing interests exist.

quintile, ever used of a computer, and frequency of listening to the radio were significant factors associated with women's attitudes justifying IPV against women (p<0.05).

## Conclusion

Over one-sixth of the respondents agreed that a husband was justified in committing IPV against women in Guyana. Public health programs focusing on geographic locations, ethnicity, and economic status must be implemented to change attitudes justifying IPV and reduce this significant public health challenge.

## Introduction

Violence against women is defined by the United Nations as "any act of gender based violence that results in, or is likely to result in physical, sexual or psychological harm or suffering to women, including threats of such acts, coercion or arbitrary deprivation of liberty, whether occurring in public or in private life" [1]. Domestic violence is a global health and societal issue, particularly in low and middle-income countries (LMICs) [2]. Domestic violence occurs within households and across a range of relationships, including couples who are married, cohabiting or dating. It can affect people, regardless of their socioeconomic and demographic backgrounds [3].

It is challenging to measure the incidence of domestic violence against women because of under-reporting due to fear of social stigma, intimidation, isolation, economic control, physical threat and harm from the perpetrators, among others [4–7]. In 2021, the World Health Organization (WHO) estimated that, globally, 1 in every 3 (30%) women experienced physical and/or sexual violence in their lifetime, mostly by an intimate partner [3]. Societal attitudes toward violence against women have been of central concern globally, and may contribute to the perpetration of domestic violence against women, and women's response to violence and victimization [3, 4]. A study by Sardinha et al. (2018), using nationally-representative data from 49 LMICs, showed that the prevalence of women's attitudes justifying the commission of violence by a husband varied from as low as 2.4% in Dominican Republic to 92.1% in Guinea [2]. At a regional level, the lowest prevalence was observed in Latin America and the Caribbean (12.0%), while South Asia as well as Sub-Saharan and North Africa had the highest prevalence (41.1% and 45.0% respectively). It was also shown that limited economic rights for women were associated with higher levels of acceptance of violence against women, while higher national female literacy rates predicted lower levels of justification [8]. In a more recent publication by the same author, using data from 161 countries and areas, it was revealed that 27% of women aged 15–49 years who have ever been in a relationship experienced intimate partner violence (IPV) in their lifetime, with wide regional disparities among countries [2]. Another analysis by Bott et al., (2019) examining data on intimate partner violence for 24 countries across the Americas showed that in Brazil, El Salvador, Panama, and Uruguay, the prevalence of women reporting ever experienced IPV in the last 12 months ranged from 7.6% in Uruguay to 58.5%. In Colombia, Guatemala, Haiti and Mexico, the prevalence seemed to decrease overtime, while in the Dominican Republic, there appeared to be in increase in the prevalence [9].

Domestic violence against women has been shown to produce short- and long-term physical, mental, sexual and reproductive health problems for women such as injuries, depression, homicide, suicide, unintended pregnancies, abortions/miscarriages, increased smoking, and substance abuse, among others [10]. Several studies have shown that geographic place of

residence, level of education, poverty, access to media, among others, to be associated with IPV against women [8, 11–13].

In Guyana, domestic violence by an intimate partner has been seen as a personal, private or family matter, and is often portrayed as justified punishment or discipline [14]. There is a scarcity of studies conducted on IPV in Guyana. However, a national survey conducted in 2018 showed that 55.0% of women who have ever had a male partner have experienced some forms of intimate partner violence during their lifetime, which is above the global average of 1 in 3 women reported by WHO in 2021 [14].

Guyana is among the countries in the Americas with the highest burden of mental health and other public health challenges [15]. Our objective was to assess women's attitudes towards the commission of IPV by a husband in Guyana, using the latest round of Multiple Indicators Cluster Surveys (MICS) conducted in 2019. To our knowledge, no previous detailed study has focused on assessing women's attitudes on IPV in Guyana. Findings from this study could help policymakers implement public health policies aimed at reducing the acceptability of all forms of physical intimate partner violence against women in Guyana.

## Materials and methods

We used data from a publicly available survey conducted in Guyana in 2019. This survey was designed to provide estimates on the situation of women aged 15 to 49 years at the national level, for urban and rural place of residence and for the ten administrative regions. In each region, the urban and rural areas were identified as the main sampling strata, and the household sample was selected in two stages. Within each stratum, a specified number of enumeration districts (EDs) were systematically selected with probability proportional to their size. Before the starting of the fieldwork, listing of the households within the EDs was carried out, and a systematic sample of 20 households was obtained from each sample ED, totalizing 435 EDs and 8,700 households for the survey. In total, 5,887 women aged 15 to 49 years were interviewed, which represented a response rate of 89.5%. The survey used standardized questionnaires to collect data on several sociodemographic, geographic location, and media access and attitude indicators for women in reproductive age (15 to 49 years). MICS surveys are conducted using Computer-Assisted Personal Interviewing (CAPI) technology. Standard procedures and programs developed under the global MICS program were adapted for the Guyana MICS6 final questionnaires, and were consistently applied. The CAPI application was pretested in urban- rural, and interior areas of regions 3 and 4 in March 2019. Based on the results of the testing phase, adjustments were made to the questionnaires and application. Prior to commencing fieldwork, all interviewers received comprehensive training on questionnaire application and data collection techniques. Team supervisors were tasked with daily oversight of fieldwork activities. Additionally, a mandatory re-interviewing process was implemented for one household within each cluster. Continuous monitoring of interviewer skills and performance was conducted on a daily basis. The survey was conducted by the Bureau of Statistics, but with technical and financial supports from UNICEF, the Inter-American Development Bank (IDB) and the Government of Guyana. Ethical approval was obtained from the Guyanese Ministry of Health-Institutional Review Board (MOH-IRB). More details on MICS can be found elsewhere [16].

According to the latest national census conducted in 2012, the country is divided into ten administrative regions (From 1 to 10) [17]. These 10 regions are divided in Coastland and Hinterland regions. The Hinterland population accounts for 10.9% of the total population, while the Coastland accounts for 89.1%. In both Coastal and Hinterlands regions, about 74% of the total population are living in rural areas. Besides, the non-indigenous majority are

concentrated in the coastal plain, while most Amerindians (Indigenous) people live in the Hinterland regions. Region 4, which includes the capital city, Georgetown, accounts for most of Guyana's administrative and economic activities [17]. In this study, we refer to the Interior location as the Hinterland region. We used both Coastal and Interior location in the study to allow for local comparisons and interpretations as the terms urban and rural are rarely used in Guyana, and urban-rural place of residence to allow for comparisons with studies conducted in other settings.

## Outcomes variables

The prevalence of women's attitudes justifying IPV against women was the main outcome we analyzed. Respondent reasons included if she: "goes out without telling him", "neglects the children", "argues with him", "refuses sex with him", "burns the food", "has another partner", "stays out late/partying", "refuses to cook or clean", "overspends", and/or "he doesn't have access to her cellphone". We also assessed women's attitude about IPV against women for any of these 10 respondent reasons.

Data on women's attitudes about IPV were collected through unprompted answers provided by women to questions: *Sometimes a husband is annoyed or angered by things that his wife does. In your opinion, is a husband right in hitting or beating his wife in the following situations: [a] If she goes out without telling him? [b] If she neglects the children? [c] If she argues with him? [d] If she refuses to have sex with him? [e] If she burns the food? [f] If she has another partner? [g] If she stays out late/partying? [h] If she refuses to cook or clean? [i] If he does not have access to her cellphone? [j] If she overspends?* Options of answers were *Yes/No or Don't know* for each question. These questions have been used for decades by MICS to assess the prevalence of women's attitudes that condone or excuse IPV against women [16].

The percentage for any of these indicators was assessed separately as well as for women who answered "Yes" for any of these 10 reasons. MICS assessed women's attitudes about IPV by asking the above questions to the women in order to capture the social justification for physical violence. In this study, women's attitudes about IPV were measured using indicators for physical violence, and did not include other forms of IPV.

## Independent variables

Several independent variables were assessed in this study. We selected these variables based on their availability in the survey, and on previous studies demonstrating that these variables were associated with women's attitudes justifying IPV against women [11, 18–20]. These variables were organized in two groups: sociodemographic, and phone ownership and access to media. The sociodemographic variables included women's age in years (less than 20, 20 to 34, 35 or more), urban-rural place of residence, coastal and interior location, regions of residence (from region 1 to region 10), ethnicity of household's head (African/Black, Amerindian, East Indian, Mixed race), and marital status (currently married/in union, formerly married/in union, never married/in union). We also assessed women's education (primary, secondary, tertiary), and wealth assets index in quintile (poorest, second, middle, fourth, richest).

Phone ownership and media access included: Own a mobile phone (yes/no), ever used of internet (yes/no), ever used of a computer or a tablet (yes/no). The frequency of listening to the radio and watching TV were categorized as "not at all", "<1 a week", "at least 1 a week", "almost every day".

The wealth asset index is pre-calculated in MICS [21]. This calculation is done through principal component analyses (PCA) using variables on household assets, building materials of the dwelling and access to utilities such as electricity, water, and sanitation. The index is

usually broken in five equally sized groups of households known as quintile where quintile 1 (Q1) represents approximately the poorest 20% of households and quintile 5 (Q5), the richest 20%.

## Statistical analyses

We carried out descriptive analyses for all the outcomes and the independent variables. We calculated the percentage of women's attitudes about IPV against women according to each independent variable using the chi-squared test. We described inequality in the percentage of each outcome by considering variables that are most frequently used for health inequality monitoring worldwide (urban-rural place of residence, ethnicity, regions of residence, wealth assets index, and the Coastal/interior location due to its local importance in Guyana) [22].

Two complex measures of inequalities were calculated to assess inequalities in the outcomes at national level. These are the slope index of inequality (SII), and the concentration index of inequality (CIX) [22]. The SII measures the absolute inequalities, and represents the differences between the fitted values for the percentage of the outcomes between the top and the bottom of the wealth scale, while taking into consideration all the other subgroups. If there is no inequality between subgroups, SII take values of zero. Positive values indicates that the outcome is more prevalent among the richest, while negative values indicate the opposite [22].

CIX measures the relative inequalities and reflects the health gradient across multiple subgroups with natural ordering. Its calculation is similar to the Gini coefficient, which is used to measure how much income is concentrated in the hands of the richest in a given country. CIX takes values from -1 to +1, where zero means absence of inequality. Positive values indicate that the outcome is more concentrated among the richest and negative values mean the opposite [22].

Multivariate logistic regression and backward stepwise variable selection were used to identify factors associated with the outcomes. To do this, for each outcome, we included in the model only those variables whose p-value is less than 5% in the raw analyses. We then used a backward procedure to exclude variables with p-value > 0.05, starting with the variables with the highest p-value [23, 24]. Statistical significances were determined with p value <5%. All the analyses were carried out using excel (version 2016) and Stata (StataCorp. 2017. Stata Statistical Software: Release 15. College Station, TX: StataCorp LLC).

## Results

S1 Table in S1 File shows the characteristics of the population studied and percentages distribution of the outcomes by selected characteristics. About 5,887 women aged 15 to 49 years were included in the study. Of these women, 75.8% lived in rural areas, 93.0% in the Coastal regions, and 43.6% in region 4. Majority of the women (47.3%) were aged between 20 to 34 years, 43.7% were East Indian, 67.8% were currently married or in union, 72.4% had a secondary level of education, and 43.0% belonged to the richer subgroup. Besides, 87.9% owned a mobile phone, 65.5% had ever used the internet, 59.1% had ever use a laptop or tablet, 28.4% and 45.2% respectively listened to radio and TV almost every day.

The prevalence of women's attitude justifying IPV against women if there was a 'yes' response to any of the 10 reasons was 17.9% (95%CI: 16.6–19.3%). "If she has another partner" was the most common reason (9.8%) reported by women for justifying IPV against women, followed by "if she stays out late/partying" (7.8%), and "if she refuses to cook or clean" (6.4%) (Web supporting information Table 1 and Fig 1). The chi-square analysis showed statistically significant differences between the outcomes with most of the independent variables (Table 1). Few exceptions were between "If she goes out without telling him" with women's age and

**Table 1. Inequalities women's attitudes towards IPV against women in Guyana (2019).**

| Indicators | Slope index of inequality (SE) | Concentration index of inequality (SE) |
|---|---|---|
| If she goes out without telling him | -5.90 (0.41) | -0.31 (0.05) |
| If she neglects the children | -8.95 (0.92) | -0.22 (0.04) |
| If she argues with him | -8.20 (0.80) | -0.31 (0.06) |
| If she refuses sex with him | -5.92 (1.74) | -0.26 (0.06) |
| If she burns the food | -4.99 (0.31) | -0.27 (0.03) |
| If she has another partner | -15.26 (1.41) | -0.23 (0.03) |
| If she stays out late/partying | -12.14 (1.83) | -0.25 (0.03) |
| If she refuses to cook or clean | -7.43 (1.73) | -0.18 (0.03) |
| If he does not have access to her cellphone | -6.79 (0.35) | -0.29 (0.04) |
| If she overspends | -5.95 (0.59) | -0.31 (0.04) |
| For any of these ten reasons | -19.49 (1.68) | -0.18 (0.02) |

frequency of listening to radio; "if she argues with him", and "if she over spends" with women's age; and "if she neglects the children" with frequency of listening to radio. No statistically significant differences were also observed between the outcomes and marital status, except for any of the 10 reasons (S1 Table in S1 File and Fig 1).

Figs 2 to 5 shows the inequality analyses by selected stratifiers. The prevalence of women's attitudes justifying IPV against women was systematically more common among women living in rural areas (Fig 2), in the Interior location (Fig 3), among the Amerindian (Fig 4), and women from the poorest household (Fig 5). A similar picture was also observed for those living in region 1 (S1 Fig in S1 File).

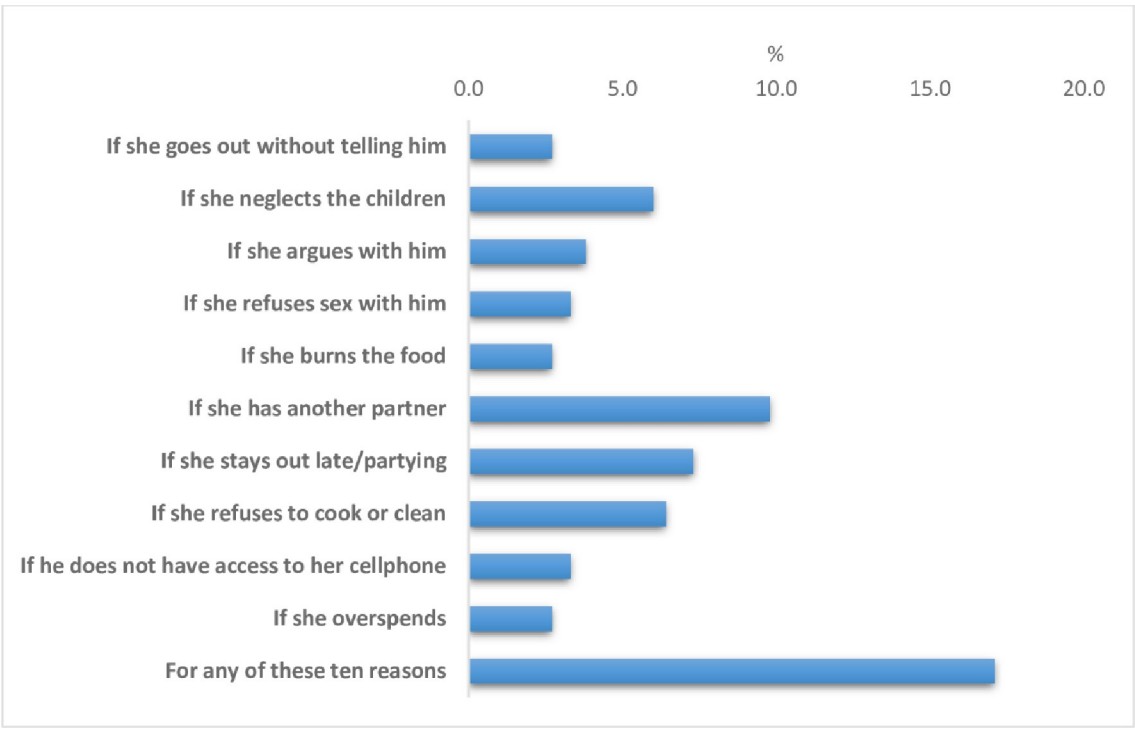

**Fig 1. Prevalence of women's attitudes towards intimate partner violence against women in Guyana (2019).**

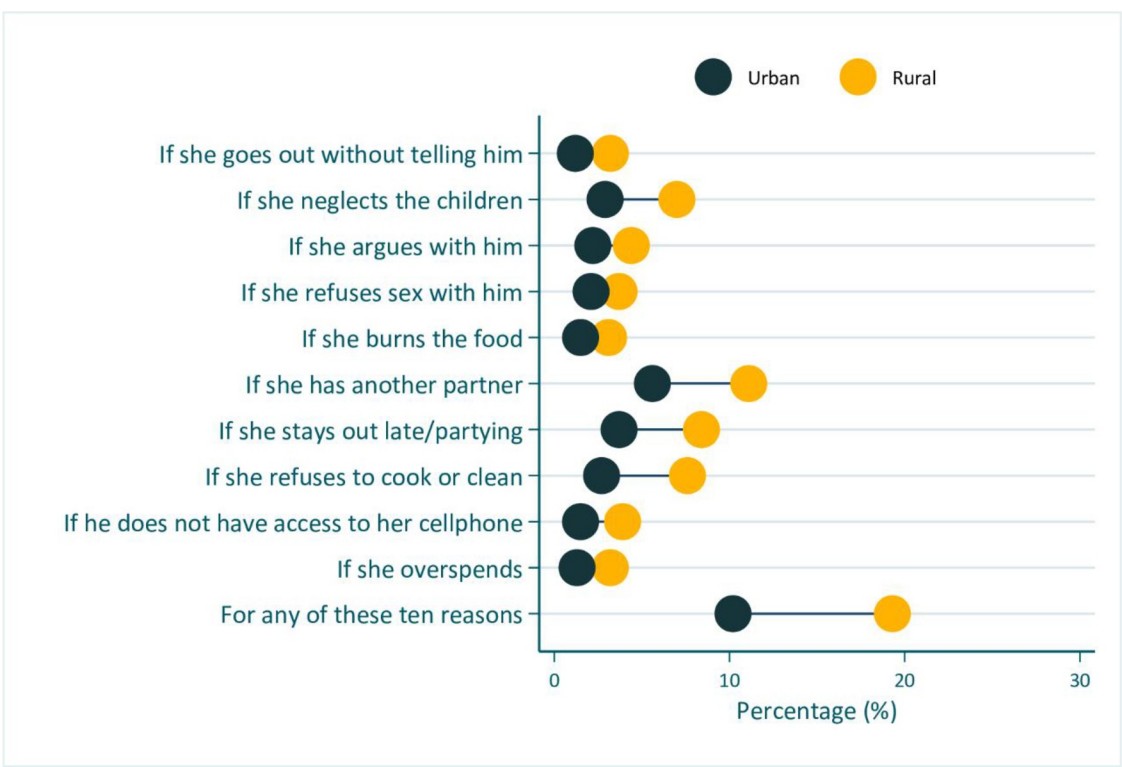

**Fig 2. Prevalence of women's attitudes towards intimate partner violence against women in Guyana (2019), by urban-rural place of residence.**

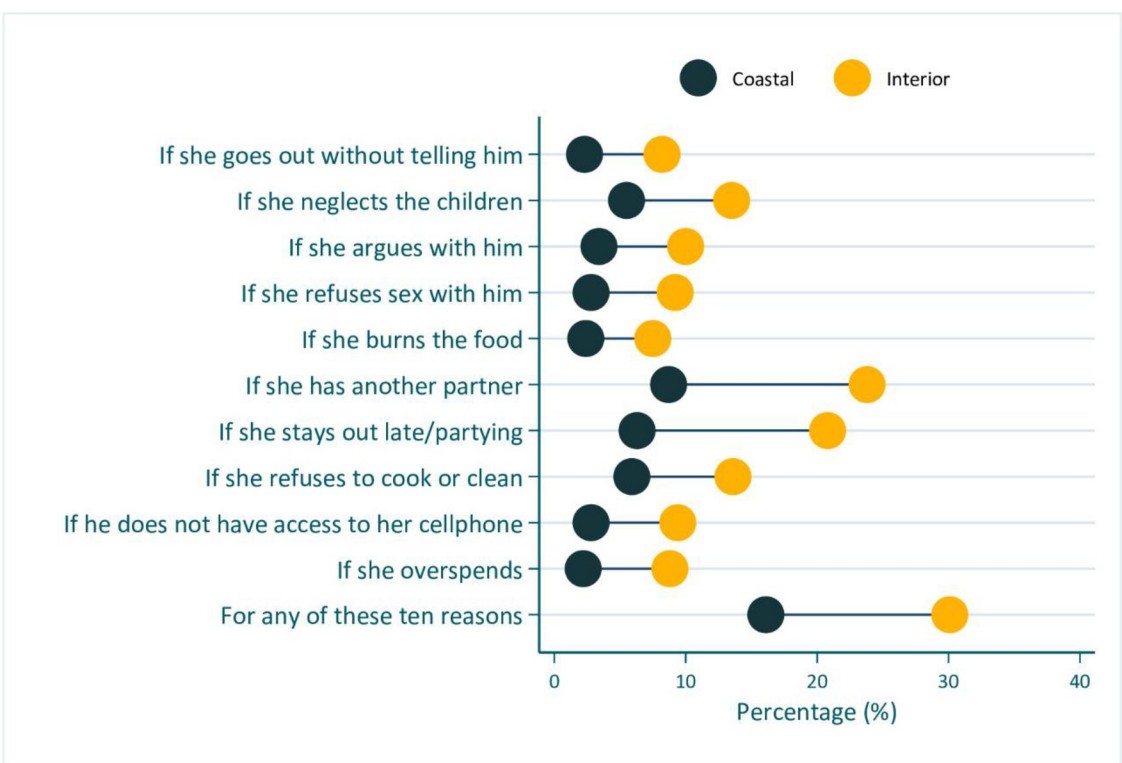

**Fig 3. Prevalence of women's attitudes towards intimate partner violence against women in Guyana (2019), by coastal-interior place of residence.**

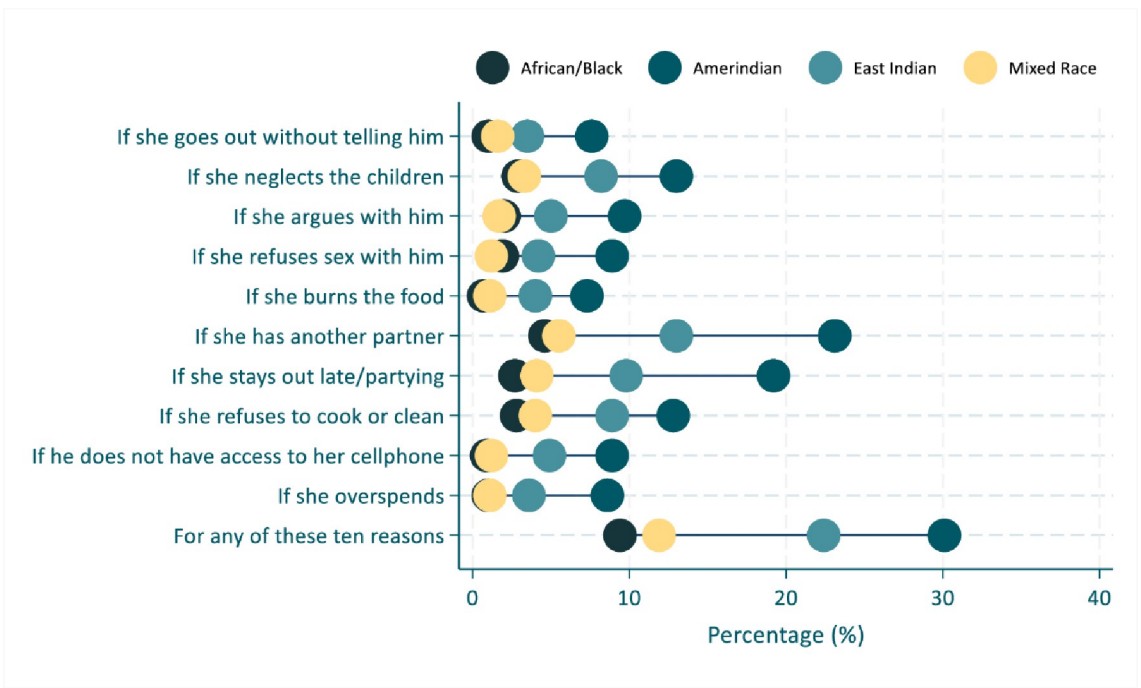

**Fig 4. Prevalence of women's attitudes towards intimate partner violence against women in Guyana (2019), by ethnic group.**

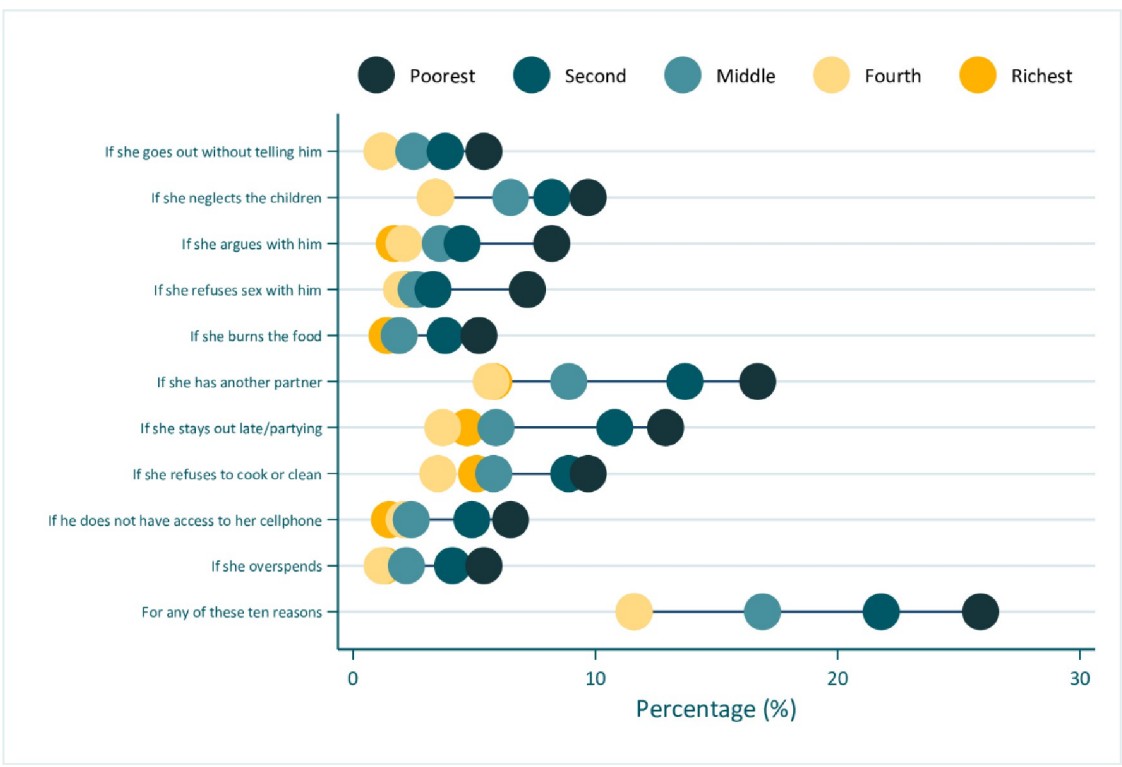

**Fig 5. Prevalence of women's attitudes towards intimate partner violence against women in Guyana (2019), by wealth quintile.**

The SII showed that women attitudes towards justifying IPV against women were more prevalent among women from the poorest household, and varied from minus 4.99 (0.31) if she burns the food to minus 15.26 (1.41) if she has another partner (Table 1). The CIX also showed that such attitudes were more concentrated among women from the poorest household, and varied from minus 0.18 (0.03) if she refuses to cook or clean to minus -0.31 (0.05) if she goes out without telling him, argues with him (-0.31 (0.06)), or overspends (-0.31 (0.04)) (Table 1).

S2 Table in S1 File presents the multivariate logistic regression analysis for the association between the covariates and women's attitudes justifying IPV against women. We excluded Coastal and Interior location from the analysis to allow comparison with studies conducted in other settings. Women who lived in rural areas (AOR: 1.67; 95%CI: 1.32–2.11), self-reported of Amerindian (AOR: 2.33; 95%CI: 1.70–3.20), or East Indian (AOR: 2.12; 95CI%: 1.63–2.77) were more likely to justify wife-beating by a husband for any of the 10 reasons. We observed a decreased of the Odds ratio according to the regions where the women lived (region 2 (AOR: 0.69; 95%CI: 0.49–0.98); region 3 (AOR: 0.67; 95%CI: 0.46–0.97); region 4 (AOR: 0.24; 95%CI: 0.16–0.36); region 5 (AOR: 0.37; 95%CI: 0.25–0.56); region 6 (AOR: 0.59; 95%CI: 0.42–0.82); region 8 (AOR: 0.46; 95%CI: 0.30–0.70); region 9 (AOR: 0.59; 95%CI: 0.41–0.85), and region 10 (AOR: 0.34; 95%CI: 0.22–0.53). There was also decrease in such attitudes for women with tertiary level of education (AOR: 0.28; 95%CI: 0.17–0.47), from the fourth (AOR: 0.51; 95%CI: 0.37–0.71), and from the richest household (AOR: 0.59; 95%CI: 0.42–0.83). If she ever used a computer or a tablet (AOR: 0.71; 95%CI: 0.57–0.89) and if she listens to radio almost every day (AOR: 0.78; 95%CI: 0.57–0.94) also were statistically associated with these attitudes (p value <0.05). When considering each component of the 10 reasons separately, statistically significant association was found with most of the independent variables, except for age of the women, marital status, and the frequency of watching TV. Geographic regions were only associated with if she "has another partner, stays out late or partying, and overspends". Likewise, ethnicity of the household's head showed statistical significant association with if she "burns the food, stays out late or partying, refuses to cook or clean", of if he does not have access to her cellphone (p value <0.05).

## Discussion

This study identified that over one-sixth of women in Guyana consider IPV against women to be a justifiable act. Such justification varied widely according to each reason assessed. Rural place of residence, ethnicity of the household's head, geographic region, level of education, wealth quintile, ever used of a computer, ever use of internet, and frequency of listening to radio were key factors associated with women's attitudes justifying the commission of violence by a husband. In addition, marked inequalities were observed between geography, ethnicity, and between the poorest and richest women.

There is lack of studies in the literature that use the combination of these ten reasons to assess women's attitudes about IPV against women. Most of the studies focuses on the first five reasons (if she: "goes out without telling him", "neglects the children", "argues with him", "refuses sex with him", "burns the food") to refer to as social justification for physical IPV against women [8, 11, 13]. Besides, it seems questions on the following reasons (if she: "has another partner", "stays out late/partying", "refuses to cook or clean", "overspends", and if "he doesn't have access to her cellphone") were only asked during the sixth round of MICS (MISC6) by a few countries, which makes it difficult the comparison of our findings with studies from other countries. However, a study conducted by Alam and Sultana in Bangladesh (2021), showed that the percentage of women aged 15–49 years who justified any of these five reasons (if she: "goes out without telling him", "neglects the children", "argues with him",

"refuses sex with him", "burns the food") was 25.5% [11]. Women's education, ethnicity and wealth index were reported to be the major contributing factors for accepting wife-beating [11]. Likewise, Tran et al. (2016), using a similar approach with MICS data from 39 low and middle-income countries, showed that the proportions of women who justified any of these five reasons varied from 2.0% in Argentina to 90.2% in Afghanistan [13]. This belief was reported to be more common among women living in rural areas, the poorest women, and those aged less than 25 years [13]. Another study by Okenwa-Emegwa et al., in Nigeria (2016) showed that almost half of Nigerian women justified any of these five reasons of wife-beating by a husband [25]. Women's age, urban-rural place of living, wealth asset index, and access to media were found to be associated with such attitudes [25]. However, in contrast to our findings, the frequency of listening to radio/TV seemed to increase the likelihood of justifying wife-beating by a husband in this country [25]. Alam et al., (2022), also revealed urban-rural place of living, level of education among women, marital status, ethnicity, wealth quintile index, and exposure to mass media as significant factors statistically associated with women's attitudes justifying IPV against women [11].

There is a lack of evidence in the literature on the attitudes of women towards justifying IPV in Guyana. However, to respond to the challenges pose by IPV against women, the Guyanese governments had taken several measures to formally support victims of intimate partner violence. The Guyanese constitution in its Article 149 declares that all Guyanese citizens have equal rights before the law regardless of ethnicity, religion, gender or place of living [26]. However, the patriarchal norms in which the social status of men is higher than that of women has been considered as key driver for violence against women in Guyana [14]. These patriarchal norms were evidenced in a study conducted by Contreras-Urbina et al. (2019), where 83.0% of the women interviewed agreed that it is natural that men should be the household's head. Besides, 78.0% of them agreed that the primary role of a woman is to take care of the home [14].

Other measures adopted by the Guyanese governments were: the 1996 Domestic violence Act, the 2009 Protection of Children Act, the 2010 Sexual Offence Act, among others [14]. Guyana is among the countries that signed and ratified the six leading international conventions that aim to protect women IPV. These are: the Convention on the Elimination of All Forms of Discrimination against Women (CEDAW); the Convention on the Rights of the Child; the International Covenant on Civil and Political Rights; the International Covenant on Social, Economic and Cultural Rights; the Convention Against Torture; and the Inter-American Convention on the Prevention, Punishment, and Eradication of Violence against Women (the Belém do Pará Convention) [14]. Despite all these legislations and conventions, it seems that living in rural areas or the Interior, being Amerindian, and belonging to the poorest segments of the population appear to reinforce the patriarchal norms that lead women to justify wife-beating against women in Guyana.

## Strengths and limitations

Our study had important strengths and limitations. We used data from MICS6 in Guyana that was designed to provide estimates that are representative at both national and regional level. MICS are international household surveys launched by the United Nations Children's Fund (UNICEF), and have been conducted in 120 LMICs since 1990 to date. We showed the magnitude of inequalities between subcategories of women as well as key factors that are associated with women's attitudes towards justifying IPV against women. Such findings have policy implications and may guide policymakers in identifying the subgroups that deserve more attention. However, because the study used data from a cross-sectional survey, we cannot

establish causal inference between the covariates and the outcomes. The reliability and validity of the instrument used by MICS to measures women's attitudes about IPV against women are still uncertain [27, 28]. However, these questions on women's attitudes about IPV against women have been used for decades to measures such attitudes in several countries [8, 11, 13, 18, 29]. Furthermore, we limited our study to physical IPV because MICS 2019 only collected data on attitudes towards physical violence, and excluded all the other forms of IPV against women. There is a lack of study in the literature assessing women's attitudes about IPV against women for any of these ten reasons, which limited the comparison of our findings with studies from other settings.

## Conclusion and recommendations

This study found that women's attitudes justifying IPV against women is high in Guyana. Marked inequalities were observed between women living in urban and rural area, between the Coastal and the Interior, between geographic regions and wealth quintile index. Rural place of residence, ethnicity of the household's head, geographic region, level of education, wealth quintile, ever used of a computer, ever use of internet, and frequency of listening to radio appeared to be the main factors associated with such attitudes. Public health programs focusing on geography, ethnicity, and economic status must be implemented to change women's attitudes about IPV against women and reducing this significant and devastating public health problem.

## Supporting information

**S1 File. Supporting information.**
(PDF)

**S1 Dataset.**
(XLSX)

## Acknowledgments

This study was carried out as part of the Guyanese Research in Injury and Trauma Training (GRITT) program to facilitate capacity building by providing in-depth training in research projects and methods and by stimulating interest and promoting knowledge in suicide, trauma, and injury prevention in Guyana.

## Author Contributions

**Conceptualization:** Gary Joseph, Charles C. Branas, Christopher N. Morrison.

**Data curation:** Gary Joseph.

**Formal analysis:** Gary Joseph.

**Investigation:** Gary Joseph.

**Methodology:** Gary Joseph.

**Project administration:** Gary Joseph.

**Resources:** Gary Joseph.

**Software:** Gary Joseph.

**Supervision:** Gary Joseph, Charles C. Branas, Christopher N. Morrison.

**Writing – original draft:** Gary Joseph.

**Writing – review & editing:** Gary Joseph, Luis Paulo Vidaletti, Cona Husbands, Lisa Edwards, Michelle K. James, Charles C. Branas, Christopher N. Morrison.

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
