## [Decision Letter · Decision Letter 0]

9 Apr 2024

PONE-D-23-38372Attitudes of women towards justifying domestic violence in Guyana: a national surveyPLOS ONE

Dear Dr. JOSEPH,

Thank you for submitting your manuscript to PLOS ONE. After careful consideration, we feel that it has merit but does not fully meet PLOS ONE’s publication criteria as it currently stands. Therefore, we invite you to submit a revised version of the manuscript that addresses the points raised during the review process.Update the Introduction and Discussion section based on the most recent papers published on this topic.The authors write “We then used a backward procedure to exclude variables with p-value > 0.2…”. It is necessary to add the justification/references for choosing 0.2.Add the justification for selecting covariates for this study.  Add the model fitting results in the Results section.Add strengths and limitations of the study ==============================

We look forward to receiving your revised manuscript.

Kind regards,

Md. Moyazzem Hossain

Academic Editor

PLOS ONE

Journal Requirements:

Reviewers' comments:

Reviewer's Responses to Questions

**Comments to the Author**

1. Is the manuscript technically sound, and do the data support the conclusions?

Reviewer #1: Yes

Reviewer #2: Yes

2. Has the statistical analysis been performed appropriately and rigorously? 

Reviewer #1: Yes

Reviewer #2: Yes

3. Have the authors made all data underlying the findings in their manuscript fully available?

Reviewer #1: Yes

Reviewer #2: Yes

4. Is the manuscript presented in an intelligible fashion and written in standard English?

Reviewer #1: Yes

Reviewer #2: Yes

5. Review Comments to the Author

Reviewer #1: It has been said about the attitude survey tool that it is standard.

Did the researcher make it? did check validity and reliability? How is the attitude measurement tool standardized??

Used its in a similar study?

In the methodology part, it is mentioned the interview with the studied samples and again in the next few lines it is mentioned the yes and no questions that check the attitude.

As a result, it is not clear for the reader that the method of data collection for attitude . interview? questionnaire?

Have you visited people's homes and collected data? Have they visited the community health centers and accessed the samples?

The method of data collection is unclear.

The type of study is not clearly stated.

analytical? Cross-sectional description?

The questions used to check the attitude seem to be the reality in married life. The fact that my wife burns food and the answer is yes or no is actually checking the behavior of my life partner.

It seems that he did not examine the attitude questions.

Is the attitude high? does not mean anything ?Attitude is usually reported as positive and negative

became .

Reviewer #2: The work is amazing and I loved to read it. Though the topic is very interesting and sensitive but the manuscript is very well written. More work is needed to ensure women right. However,Congratulations to all the authors.

6. PLOS authors have the option to publish the peer review history of their article (what does this mean?). If published, this will include your full peer review and any attached files.

Reviewer #1: No

Reviewer #2: No

---

## [Author Response · Author response to Decision Letter 0]

30 Apr 2024

Response to reviewers

Update the Introduction and Discussion section based on the most recent papers published on this topic.

Reply: We thank the reviewers for this comment. We have reviewed the citations provided in both the Introduction and Discussion section, and updated the citations based on the most recent papers published on this topic.

The authors write “We then used a backward procedure to exclude variables with p-value > 0.2…”. It is necessary to add the justification/references for choosing 0.2.

Reply: We thank the reviewers for this comment. However, there was an error in the write-up. Although this cut-off can be utilized in a regression model as per the literature, in our analysis, we omitted variables with a p-value > 0.05. We have made the necessary revisions and included references to support our approach. 

Add the justification for selecting covariates for this study. 

Reply: We thank the reviewer for this comment. We now provide justification for selecting the covariates used in the study.

“Several independent variables were assessed in this study. We selected these variables based on their availability in the survey, and on previous studies demonstrating that these variables were associated with women’s attitudes justifying IPV against women”

Add the model fitting results in the Results section.

Reply: Done

Add strengths and limitations of the study

Reply: Done

Strengths and limitations

Our study had strengths and limitations. We used data from MICS6 in Guyana that was designed to provide estimates that are representative at both national and regional level. MICS are international household surveys launched by the United Nations Children’s Fund (UNICEF), and have been conducted in 120 LMICs since 1990 to date. We showed the magnitude of inequalities between subcategories of women as well as key factors that are associated with women’s attitudes towards justifying IPV against women. Such findings have policy implications and may guide policymakers in identifying the subgroups that deserve more attention. However, because the study used data from a cross-sectional survey, we cannot establish causal inference between the covariates and the outcomes. The reliability and validity of the instrument used by MICS to measures women’s attitudes about IPV against women are still uncertain. However, these questions on women’s attitudes about IPV against women have been used for decades to measures such attitudes in several countries. Furthermore, we limited our study to physical IPV because MICS 2019 only collected data on attitudes towards physical violence, and excluded all the other forms of IPV against women. There is a lack of study in the literature assessing women’s attitudes about IPV against women for any of these ten reasons, which limited the comparison of our findings with studies from other settings.

We note that Figure 1 in your submission contain [map/satellite] images which may be copyrighted. All PLOS content is published under the Creative Commons Attribution License (CC BY 4.0), which means that the manuscript, images, and Supporting Information files will be freely available online, and any third party is permitted to access, download, copy, distribute, and use these materials in any way, even commercially, with proper attribution. For these reasons, we cannot publish previously copyrighted maps or satellite images created using proprietary data, such as Google software (Google Maps, Street View, and Earth). For more information, see our copyright guidelines: http://journals.plos.org/plosone/s/licenses-and-copyright.

 We require you to either (1) present written permission from the copyright holder to publish these figures specifically under the CC BY 4.0 license, or (2) remove the figures from your submission

Reply: We thank the editorial team for this remark. We decided to remove figure 1 from our manuscript to adhere to PLOS ONE copyright guidelines on Figures publication.

Reviewer #1: It has been said about the attitude survey tool that it is standard.

Did the researcher make it? did check validity and reliability? How is the attitude measurement tool standardized? Used its in a similar study?

Reply: We appreciate the reviewer's comment. However, it's important to clarify that we specified in the Materials and Methods section that our study utilized secondary data from MICS, and we did not conduct the survey ourselves. As for the standardization of the survey tool, we acknowledged it as a limitation of our study. Nevertheless, the questions employed to assess women's attitudes towards wife-beating by a husband have been utilized for decades in over 120 countries for this purpose. Additionally, all the questions underwent pre-testing before the commencement of the fieldwork.

In the methodology part, it is mentioned the interview with the studied samples and again in the next few lines it is mentioned the yes and no questions that check the attitude.

As a result, it is not clear for the reader that the method of data collection for attitude . interview? questionnaire?

Reply: We thank the reviewer for this comment. The methodology used by MICS to collect the data are now provided, and with reference.

“MICS surveys are conducted using Computer-Assisted Personal Interviewing (CAPI) technology. Standard procedures and programs developed under the global MICS program were adapted for the Guyana MICS6 final questionnaires, and were consistently applied. The CAPI application was pre-tested in urban- rural, and interior areas of regions 3 and 4 in March 2019. Based on the results of the testing phase, adjustments were made to the questionnaires and application. Prior to commencing fieldwork, all interviewers received comprehensive training on questionnaire application and data collection techniques. Team supervisors were tasked with daily oversight of fieldwork activities. Additionally, a mandatory re-interviewing process was implemented for one household within each cluster. Continuous monitoring of interviewer skills and performance was conducted on a daily basis”.

Have you visited people's homes and collected data? Have they visited the community health centers and accessed the samples?

Reply: We thank the reviewer for this comment. As explained earlier, we only used secondary data from MICS to conduct this study.

“We used data from a publicly available survey conducted in Guyana in 2019. This survey was designed to provide estimates on the situation of women aged 15 to 49 years at the national level, for urban and rural place of residence and for the ten administrative regions”.

The method of data collection is unclear.

Reply: We appreciate the reviewer’s comment. More details are now provided on the method of data collection as explain earlier.

“We used data from a publicly available survey conducted in Guyana in 2019. This survey was designed to provide estimates on the situation of women aged 15 to 49 years at the national level, for urban and rural place of residence and for the ten administrative regions. In each region, the urban and rural areas were identified as the main sampling strata, and the household sample was selected in two stages. Within each stratum, a specified number of enumeration districts (EDs) were systematically selected with probability proportional to their size. Before the starting of the fieldwork, listing of the households within the EDs was carried out, and a systematic sample of 20 households was obtained from each sample ED, totalizing 435 EDs and 8,700 households for the survey. In total, 5,887 women aged 15 to 49 years were interviewed, which represented a response rate of 89.5%. The survey used standardized questionnaires to collect data on several sociodemographic, geographic location, and media access and attitude indicators for women in reproductive age (15 to 49 years). MICS surveys are conducted using Computer-Assisted Personal Interviewing (CAPI) technology. Standard procedures and programs developed under the global MICS program were adapted for the Guyana MICS6 final questionnaires, and were consistently applied. The CAPI application was pre-tested in urban- rural, and interior areas of regions 3 and 4 in March 2019. Based on the results of the testing phase, adjustments were made to the questionnaires and application. Prior to commencing fieldwork, all interviewers received comprehensive training on questionnaire application and data collection techniques. Team supervisors were tasked with daily oversight of fieldwork activities. Additionally, a mandatory re-interviewing process was implemented for one household within each cluster. Continuous monitoring of interviewer skills and performance was conducted on a daily basis”.

The type of study is not clearly stated. analytical? Cross-sectional description?

Reply: We thank the reviewer for this comment. The study design is now provide in the Title of the study.

“Attitudes of women towards intimate partner violence in Guyana: a cross-sectional analytical study”.

The questions used to check the attitude seem to be the reality in married life. The fact that my wife burns food and the answer is yes or no is actually checking the behavior of my life partner.

Reply: We appreciate the reviewer's comment. It's worth noting that these questions are standardized by both MICS and the Demographic Health Survey (DHS) to measure such attitudes among both men and women across more than 200 countries. However, we acknowledge this aspect as a limitation of our findings.

It seems that he did not examine the attitude questions.

Reply: We thank the reviewer for this comment. However, we used the exact same questions used by MICS to assess women’s attitudes about intimate partner violence to conduct this study.

Is the attitude high? does not mean anything ?Attitude is usually reported as positive and negative became .

Reply: We appreciate the reviewer's comment. While we acknowledge that there are various methods to measure an outcome variable, including different question formats, it's important to recognize that the Yes/No questions used by MICS to assess women's attitudes towards justifying intimate partner violence against women in Guyana, is being used in more than 120 to assess such attitudes. Additionally, the option of providing both positive and negative responses can be comparable to a Yes/No format. Furthermore, it's worth noting that these questions underwent pre-testing in Guyana before the commencement of the fieldwork. We hope that these questions will be validated in the near future.

---

## [Decision Letter · Decision Letter 1]

3 May 2024

Attitudes of women towards intimate partner violence in Guyana: a cross-sectional analytical study

PONE-D-23-38372R1

Dear Dr. JOSEPH,

We’re pleased to inform you that your manuscript has been judged scientifically suitable for publication and will be formally accepted for publication once it meets all outstanding technical requirements.

Kind regards,

Md. Moyazzem Hossain

Academic Editor

PLOS ONE

Additional Editor Comments (optional):

Reviewers' comments:

Reviewer's Responses to Questions

**Comments to the Author**

1. If the authors have adequately addressed your comments raised in a previous round of review and you feel that this manuscript is now acceptable for publication, you may indicate that here to bypass the “Comments to the Author” section, enter your conflict of interest statement in the “Confidential to Editor” section, and submit your "Accept" recommendation.

Reviewer #1: All comments have been addressed

2. Is the manuscript technically sound, and do the data support the conclusions?

Reviewer #1: Yes

3. Has the statistical analysis been performed appropriately and rigorously? 

Reviewer #1: Yes

4. Have the authors made all data underlying the findings in their manuscript fully available?

Reviewer #1: Yes

5. Is the manuscript presented in an intelligible fashion and written in standard English?

Reviewer #1: Yes

6. Review Comments to the Author

Reviewer #1: The authors have carefully made all the necessary corrections and provided a complete explanation.

These amendments have been accepted and I have no special opinion.

1. The study presents the results of original research. ok

2. Results reported have not been published elsewhere.ok

3. Experiments, statistics, and other analyses are performed to a high technical standard and are described in sufficient detail.ok

4. Conclusions are presented in an appropriate fashion and are supported by the data.ok

5. The article is presented in an intelligible fashion and is written in standard English.ok

6. The research meets all applicable standards for the ethics of experimentation and research integrity.ok

7. The article adheres to appropriate reporting guidelines and community standards for data availability.ok

7. PLOS authors have the option to publish the peer review history of their article (what does this mean?). If published, this will include your full peer review and any attached files.

Reviewer #1: No

---

## [Editor Report · Acceptance letter]

9 May 2024

PONE-D-23-38372R1 

PLOS ONE

Dear Dr. Joseph, 

I'm pleased to inform you that your manuscript has been deemed suitable for publication in PLOS ONE. Congratulations! Your manuscript is now being handed over to our production team.

Kind regards, 

on behalf of

Professor Md. Moyazzem Hossain 

Academic Editor

PLOS ONE